# Ontogenetic shape trajectory of *Trichomycterus areolatus* varies in response to water velocity environment

**Peter C. Searle**[1]*, **Margaret Mercer**[1], **Evelyn Habit**[2], **Mark C. Belk**[1]

**1** Department of Biology, Brigham Young University, Provo, Utah, United States of America, **2** Facultad de Ciencias Ambientales y Centro EULA, Departamento de Sistemas Acuáticos, Universidad de Concepción, Concepción, Chile

* petersearle94@gmail.com

**Data Availability Statement:** The data set used for analysis has been uploaded to Dryad. The public DOI is doi:10.5061/dryad.rn8pk0p8f and URL is https://doi.org/10.5061/dryad.rn8pk0p8f.

## Abstract

Body and head shape among fishes both vary between environments influenced by water velocity and across ontogeny. Although the shape changes associated with variation in average water velocity and ontogeny are well documented, few studies have tested for the interaction between these two variables (i.e., does ontogenetic shape variation differ between velocity environments). We use geometric morphometrics to characterize shape differences in *Trichomycterus areolatus*, a freshwater catfish found in high and low-velocity environments in Chile. We identify a significant interaction between velocity environment and body size (i.e., ontogeny). Ontogenetic patterns of shape change are consistent with other studies, but velocity environment differentially affects the ontogenetic trajectory of shape development in *T. areolatus*. Shape change over ontogeny appears more constrained in high-velocity environments compared to low-velocity environments.

## Introduction

Morphometric traits in animals and plants are shaped by the selective pressures of their environments resulting in interspecific variation in shape [1–4]. Shape responds sensitively to environmental pressures such as predation, resource use, competition, temperature, water velocity, and water availability [5–10]. For example, head and body size changed in two Australian snakes (*Pseudechis porphyriacus* and *Dendrelaphis punctulatus)* following introduction of a toxic cane toad (*Bufo marinus*), and head shape changed among species in the Neotropical cichlid genus *Geophagus* depending on prey availability [11,12]. Organisms vary in form (i.e., shape) among species largely in response to variation in habitat use or trophic niche and corresponding environmental variation among habitats [2,12].

Environmental variation can also lead to intraspecific morphometric differentiation [13–17]. Examples include differences in head shape in wall lizards (*Podarcis bocagei*) between saxicolous and ground-dwelling habitats; wing morphology in speckled wood butterflies (*Parage aegeria*) along a latitudinal gradient; tail, head and ear length in West African Dwarf goats (*Capra aegagrus hircus*) among agro-ecological zones and swimming performance and shape

**Funding:** Funding for this work was supported by grants to MCB by the Roger and Victoria Sant Foundation and the Department of Biology, Brigham Young University, as well as EH by projects 032904 3/R Dirección de Investigación, Universidad del Bío-Bío and 204.310.041-1.0 Dirección de Investigación, Universidad de Concepción. The funders had no role in study design, data collection and analysis, decision to publish, or preparation of the manuscript.

**Competing interests:** The authors have declared that no competing interests exist.

in Creek Chub (*Semotilus atromaculatus*) due to urbanization [18–21]. Such intraspecific phenotypic variation can be the result of genetic [22] or plastic variation [23–25].

Stream zonation is an outgrowth of the river continuum concept, which suggests that as streams move from their headwaters to their mouths, they differ in many environmental variables [26]. Across a stream continuum (i.e., across zones) there are multiple environmental variables that vary including: substrate size, stream width, depth, flow volume, gradient, and water velocity, as well as biotic variables such as prey type and availability [27]. Each of these variables may contribute to shape differences observed between fish from different stream zones. Of these variables, water velocity has been demonstrated as a strong predictor of intraspecific variation in shape in fishes [28]. Typically, high-velocity environments favor a narrower body shape, whereas low-velocity environments favor a more robust body shape [28]. This general response has been well documented in many fishes including centrarchids [29], characids [30], cichlids [31], cyprinids [32] and salmonids [33]. Although velocity is typically considered the primary selective agent in these studies, it is often difficult to separate water velocity from the other, usually covarying, variables that differ between stream zones.

In addition to variation in shape in response to the environment, an organism's shape changes over the course of development [34,35]. Since D'Arcy Wentworth Thompson's influential book, *On Growth and Form* [36], ontogenetic shape change has been well documented among invertebrates: crustaceans [37] and insects [38]; as well as vertebrates: reptiles [39], birds [40] and mammals [41]. Most fishes hatch or are born several orders of magnitude smaller in size compared to adults. Because of this dramatic change in body size as they grow, fish occupy multiple niches over the course of their ontogenetic development, and they can experience different selective pressures within each sequential niche [42]. Often associated with these ontogenetic niche shifts are similarly dramatic ontogenetic shape changes [43–45]. Generally, juveniles have larger eyes and heads relative to body size; whereas, adults have smaller eyes and heads relative to body size [46,47]. Studies from numerous fish taxa have documented similar shape changes [43–45,47,48]. Although many studies have addressed the change in form over developmental stages in organisms with reference to allometry [46,49], few studies have addressed how differing selective environments interact with ontogenetic patterns to produce phenotypic variation within species [50].

The potential interaction between environment and ontogenetic shape trajectories in fishes has been quantified in relatively few studies. In the sexually dimorphic livebearer, *Brachyraphis rhabdophora*, males exhibited parallel ontogenetic shape trajectories such that shape differences between juveniles and adults were maintained between predation environments. However, females exhibited convergent ontogenetic shape trajectories such that shape differences were more pronounced in juvenile females compared to adult females between predation environments [51]. Similarly, three-spined sticklebacks (*Gasterosteus aculeatus*) exhibited differential ontogenetic shape trajectories between two marine (with different habitat structure) and one freshwater habitat resulting in morphologically distinct adults [52] (Also see [53–58]). Taken together, these studies provide evidence that the interaction between environmental variation and ontogeny could result in differential shape trajectories within species. However, few studies have specifically quantified the interaction between water velocity environment across the ontogenetic shape trajectory in fishes.

In this paper we characterize shape variation in *Trichomycterus areolatus* across ontogeny in high-velocity and low-velocity stream environments. *Trichomycterus areolatus* is a freshwater catfish found throughout Central Chile. It is a small (up to 140 mm standard length), generalist, benthic carnivore that feeds mostly on insect larvae and crustaceans [59–61]. *Trichomycterus areolatus* is a good candidate for studying the potential interactive effect of water velocity environment on ontogenetic shape trajectories because they are one of the most

broadly distributed species in Chilean rivers and streams, and they occur in all water velocity environments (i.e., stream zones) [26]. Furthermore, all size classes of *T. areolatus* are found in all stream zones, they appear to occupy the same stream section throughout their entire lives, and there is no evidence of migration among stream sections [27]. To investigate the potential interaction between environments characterized by differences in water velocity and ontogeny (i.e. do ontogenetic shape trajectories differ between high and low-velocity environments), we used landmark-based geometric morphometrics [62] to quantify body and head shape across ontogenetic trajectories of *T. areolatus* in contrasting velocity environments. We document an interaction between velocity environment and ontogenetic shape variation in *T. areolatus*.

## Materials & methods

### Study site and collection

The Andalién River (Concepción province, Chile, 36.740673 S, 73.016947 W) is a small coastal drainage (48 km) that lacks geographic or reproductive barriers to gene flow (i.e., no waterfalls or other flow barriers) along its entire length. In addition, the river exhibits strong zonation patterns in habitat structure and environmental variables. The river can be divided into three distinct zones: the rhithron (higher gradient stream characterized by high current velocity, large substrate, and small stream size), the transition (with intermediate characteristics), and the potamon (lower gradient river characterized by low current velocity, small substrate and large stream size) [27]. We recognize that these stream zones differ in several ways (Table 1, modified from [26]), but these variables covary with stream velocity in predictable ways. Water velocity is most different between the transition and potamon zones, thus we group the rhithron and transition zones as a high-velocity environment, and the potamon zone as a low-velocity environment. As in many studies where a continuous environment is divided into discrete segments for comparison [15,63,64], there are several other variables that potentially differ between high and low-velocity environments and that could create selective differences between environments. For this reason, we refer to these as water velocity environments and we acknowledge that other stream variables may contribute to observed variation in shape in *T. areolatus*.

We collected *T. areolatus* from the Andalién River and two of its tributary streams (Nonguén and Queule Rivers) in 2003 and 2004 using electrofishing backpack equipment. All collections were done under the auspices of Dirección de Investigación, Universidad de Concepción and the Undersecretariat of Fisheries (Collections were made before IACUC protocols were required for field studies in Chile, but we followed the Guidelines for Use of Fishes in

**Table 1. Environmental characteristics of the rithron, transition, and potamon zones in the Andalién River, Chile during the wet season.**

| Parameter | Unit | Rithron | Transition | Potamon |
|---|---|---|---|---|
| Mean Depth | Cm | 26.5 ± 5.4 | 28.3 ± 11.1 | 25.5 ± 7.3 |
| Maximum Depth | Cm | 50.0 ± 6.3 | 61.8 ± 27.7 | 55.2 ± 17.4 |
| Flow | m³/s | 0.7 ± 0.6 | 3.3 ± 2.6 | 3.6 ± 2.7 |
| Avg. Velocity | m/s¹ | 0.83 ± 0.06 | 1.0 ± 0.24 | 0.59 ± 0.3 |
| Boulders | % | 76.7 ± 2.5 | 16.6 ± 25.8 | 0 ± 0 |
| Width | M | 5.8 ± 2.2 | 11.8 ±6.3 | 19.8 ± 15.8 |
| Order | | 3.3 ± 0.5 | 3.6 ± 0.5 | 4.3 ± 0.2 |
| Gradient | ° | 0.4 ± 0.09 | 0.2 ± 0.07 | 0.14 ± 0.03 |

Modified from [26].

Field Research provided by the American Society of Ichthyologists and Herpetologists, the American Fisheries Society and the American Institute of Fisheries Research Biologists [65]). Specimens were originally collected for a study comparing distribution patterns of fish assemblages in the Andalién River drainage [27]. According to guidelines in [65], we retained specimens based on population density at each location. The majority of fish were released, but a representative sample of the entire size range of the specimens was randomly retained as museum vouchers. It is these specimens retained as vouchers that form the basis of this study. Because the density of *T. areolatus* in the rithron zone was low, fewer specimens were retained (n = 20). Because these specimens from the rithron did not represent the full range of body sizes available in the other two zones, we combined samples from the rithron zone and the transition zone. In addition, water velocities in the rithron and transition zones were higher and more similar to each other than velocities in the potamon zone (Table 1). To determine if the inclusion of specimens collected in the rithron changed the statistical outcome or interpretation of the results, we ran the overall shape analysis with and without the samples from the rithron. Neither the statistical significance, nor the estimated least squares means of shape differed between the two analyses, so we report results from analysis of the full data set.

*Trichomycterus areolatus* is sexually monomorphic [66], so we did not distinguish between males and females in our sampling and analysis. We euthanized specimens with an overdose of BZ-20 (20% ethyl p-aminobenzoate) and assigned ID numbers. We preserved specimens in ethyl alcohol and measured and photographed them at Brigham Young University. We used a Canon EOS Digital Rebel XT DSLR camera with a Canon EF-S 18–55 mm f3.5–5.6 lens. Some specimens were damaged or preserved in unnatural positions such that they could not be included in the analysis. Mean sizes and size distributions were similar between high and low-velocity environments. Specimens used for the analysis of lateral body shape from high-velocity environments (n = 183) had a mean standard length of 55.2 mm (SD = 19.5 mm; range = 22.4–116.3 mm); and specimens from low-velocity environments (n = 103) had a mean standard length of 58.3 mm (SD = 23.5 mm; range = 23.5–115.1 mm). Specimens used for the analysis of head shape from high-velocity environments (n = 182) had a mean standard length of 58 mm (SD = 22 mm; range = 22.4–134.9 mm); and specimens from low-velocity environments (n = 113) had a mean standard length of 58.6 (SD = 24.3 mm; range = 23.5–138.8 mm). We photographed the right lateral view of the body, and the dorsal view of the head of each fish for morphometric analysis and included a mm-scale ruler in the photograph for scaling. Preserved specimens are deposited in the Monte L. Bean Life Science Museum at Brigham Young University in Provo, Utah, USA.

## Geometric morphometrics

We used landmark-based geometric morphometrics to quantify body and head shape in *T. areolatus* [67]. Prior to landmarking, we meticulously reviewed digital images to confirm that each specimen was not rotated dorsally or ventrally, was straight on the dorso-ventral and anteroposterior axis, had a closed mouth, and the operculum was in a relaxed, closed position. Where possible we re-photographed specimens, but if the error could not be corrected, we removed the specimen from the analysis. Most specimens were included in both the body and head shape analysis, but a few were used in only one or the other analysis. We digitized landmarks and semi-landmarks using the program tpsDig [68]. One researcher landmarked all specimens in random order without reference to collection location (i.e., velocity environment) or body size. Two other researchers independently inspected landmarked images to confirm homologous and consistent placement of landmarks. For analysis of lateral body shape, we used 13 landmarks defined as follows: 1) rostral-most point, 2) center of eye, 3)

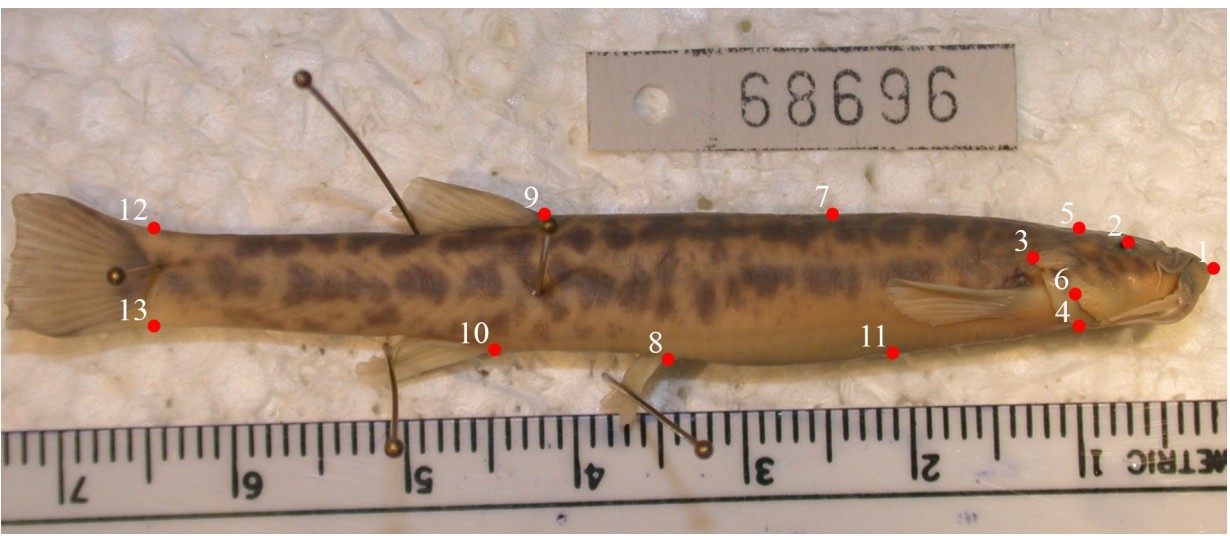

**Fig 1. Body landmarks.** Landmarks used in the analysis of shape of the lateral body view of *T. areolatus*.

posterior tip of operculum, 4) juncture of the ventral margin of the operculum with the ventral outline of the body, 5) vertical projection of the x-value of landmark 4 on the dorsal outline, 6) posterior tip of spine on preoperculum, 7) midpoint between landmarks 5 and 9 on the dorsal outline, 8) anterior origin of pelvic fin, 9) anterior origin of dorsal fin, 10) anterior origin of anal fin, 11) midpoint between landmarks 4 and 8 on the ventral outline, 12) dorsal projection from anterior extent of caudal fin, and 13) ventral projection from anterior extent of caudal fin. Landmarks 7 and 11 are sliding semilandmarks (Fig 1).

The dorsal view of the head represents a symmetrical structure, so we landmarked only the right half for analysis [69]. We used 7 landmarks defined as follows: 1) anterior extent of pre-maxilla at midline of head, 2) midline of head projected from point 7, 3) lateral edge of nostril, 4) medial margin of eye, 5) lateral margin of eye, 6) lateral extent of the head projected from point 5, and 7) posterior tip of opercular spine (Fig 2).

We generated shape variables from our landmark data using tpsRelW [70]. tpsRelW uses a generalized Procrustes analysis to remove non-shape variation (i.e., position, orientation, and scale) [71,72], and generates shape variables in the form of partial warps and uniform components (i.e., **W**, the weight matrix). The program then calculates a series of relative warps that result from a principal components analysis of the weight matrix [62,67,73]. We used these relative warps as our shape variables for analysis. In addition, we used centroid size (a multivariate measure of size) derived from this analysis to represent the stage of ontogenetic development. We excluded relative warps that individually accounted for < 1% of shape variation to avoid inflating the real degrees of freedom associated with analysis of shape variation [17,74]. Consequently, we used the first 12 of 22 relative warps for the body (accounting for 96.15% of total shape variation) and the first 9 of 10 relative warps for the head (accounting for 99.91% of total shape variation).

## Statistical analysis

To determine the effects of velocity environment and ontogeny on shape variation in *T. areolatus*, we used a multivariate linear mixed model [16,17,74]. We analyzed body and head shape separately, using relative warps as the response variable. We used location (rithron/transition versus potamon) as a discrete predictor in the model to represent either high or low-velocity

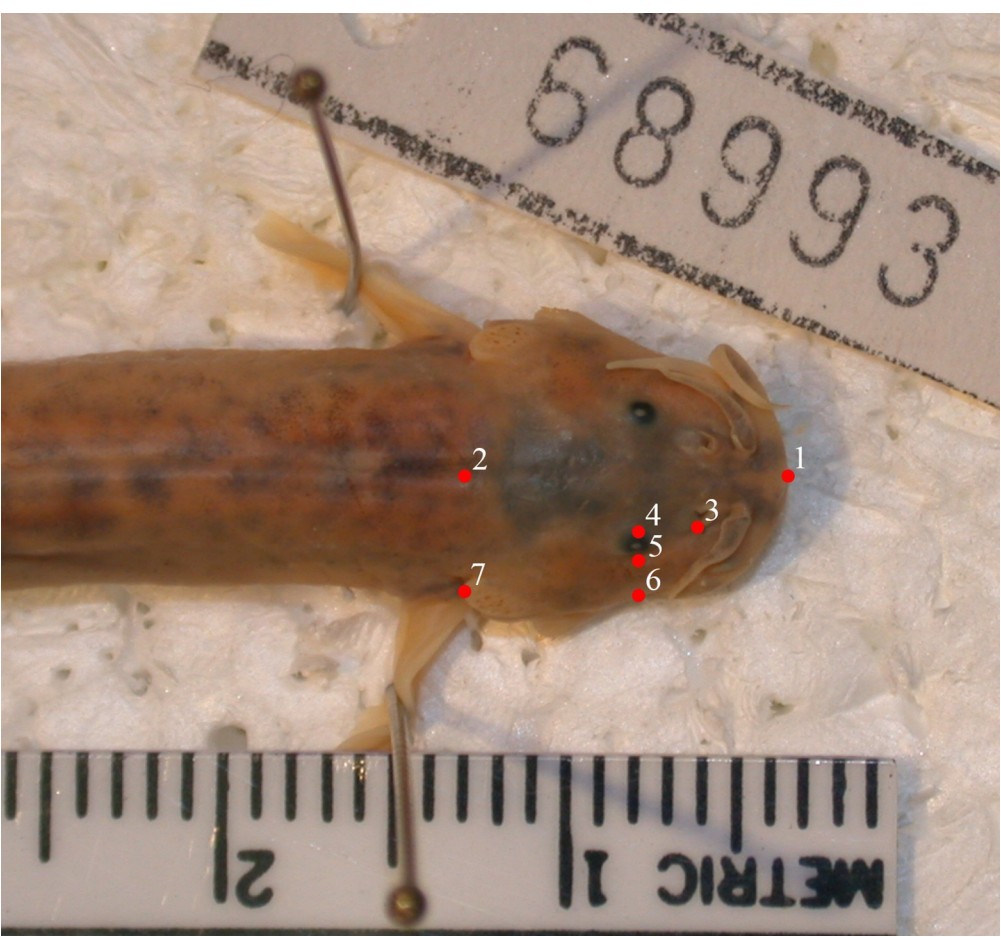

**Fig 2. Head landmarks.** Landmarks used in the analysis of shape of the head of *T. areolatus*.

environments, and we used centroid size as a covariate predictor in the model to represent ontogenetic stage of development. We included the interaction between velocity environment and centroid size (ontogeny) to test for different patterns of shape variation across ontogeny between velocity environments. The multivariate linear mixed model required that the matrix of shape variables (relative warps) be vectorized such that each individual specimen was represented by 12 rows (body) or 9 rows (head) of data, and each row corresponded to one relative warp [74]. Thus, the response variable is transposed from a row in a matrix to a vector column of responses. This vectorization creates the need for an index variable that preserves the identity of specific relative warps. This index variable is used as a predictor variable in the model much like time is used as a predictor in a typical repeated measures design [15,74–76]. Relative warps are principal components, and as such they are orthogonal to each other, they have a mean of zero, and their ordination is arbitrary relative to each other. For example, individuals that would have a positive score on one component would not necessarily score positively on any other component. For this reason, the index variable, by preserving the identity of each relative warp, is necessary to test our hypotheses of shape differences among groups. Thus, it is the two-way interactions between velocity environment and the index variable, and centroid size and the index variable, and the three-way interaction between velocity environment, centroid size, and the index variable that allow us to test the hypothesis of significant shape differences between velocity environments across ontogeny on all relative warps simultaneously

[51,74,75,77]. We included individual ID number as a random effect in our analysis because each specimen had multiple shape variables (i.e., relative warps) as response variables. We estimated degrees of freedom using the Kenward–Roger method [78] and conducted analyses using Proc MIXED in SAS (SAS version 9.4, SAS Institute Inc., Cary NC, USA). To determine which relative warps differed across ontogeny and between velocity environments, we plotted least squares means and associated 95% confidence intervals for small and large individuals separately in both velocity environments.

To quantify the magnitude and direction of shape change between velocity environments over ontogeny we used a phenotypic change vector analysis (PCVA) [79–81]. Because PCVA requires two categorical variables to form the two vectors that are compared, we used size at maturity (51 mm SL) [66] to divide the sample between juveniles and adults, and then divided the adults into small adults and large adults by dividing the size range of adults in half (adults ranged in size from 51mm to 139 mm, dividing line between small adults and large adults = 103 mm SL) [66]. We used juveniles as the small size class and large adults as the large size class to characterize the PCVA across the entire size range. The PCVA tests for differences in the magnitude and/or direction of ontogenetic shape change between velocity environments across all relative warps used in the analysis. We tested for significant differences in magnitude and direction of shape change between small and large individuals in different velocity environments using ASReml-R version 4 [82] using modified R [83] scripts from [74,84].

To visualize shape variation between velocity environments at small and large sizes we calculated divergence vectors as described in [85,86]. These divergence vectors characterize differences in shape across all relative warps between velocity environments for small and large size classes separately. We calculated the divergence vector as the sum of the products of the first eigenvector (from a principal components analysis of the least squares means for each relative warp in the two velocity environments) times the associated relative warp scores for each individual. We then used these divergence scores for each individual as a regressor on shape in tpsRegr [87] to generate a thin-plate spline visualization of the extremes of shape variation between velocity environments. These thin-plate spline plots of shape deformation represent shape divergence across all relative warps between velocity environments. Data used for all analyses are available in the Dryad Digital Repository (doi:10.5061/dryad.rn8pk0p8f).

## Results

Body shape of *T. areolatus* (right lateral view) differed significantly by centroid size (over ontogeny), and between velocity environments, and centroid size and velocity environment exhibited a significant interaction (see the two-way and three-way interaction with the index variable in Table 2). Ontogenetic shape change in the body was mainly associated with relative warp 1, with minor contributions from relative warps 4, 5, and 6. Shape differences associated with velocity environment were mainly evident on relative warp 1, with minor contributions from relative warps 5, 6, 8, and 10 (Fig 3). The magnitude of shape change of the body (p = 0.0024), but not the direction (p = 0.3889) differed significantly between velocity environments. Fish from the high-velocity environment had a significantly lower magnitude of shape change over ontogeny compared to fish from the low-velocity environment ($D_{High}$ = 0.0339, $D_{Low}$ = 0.0494,). Correspondingly, the change in shape over ontogeny in the high-velocity environment mainly involves a slight reduction in the proportional length of the head relative to the body and a slight deepening of the body in the midsection (Fig 4, right side). In contrast, the change in shape over ontogeny in the low-velocity environment includes a substantial reduction in the proportional length of the head relative to the body and a substantial narrowing of the body in the midsection (Fig 4, left side). In addition, shape differs between high-velocity and low-velocity environments

**Table 2. Multivariate analysis of covariance effects for body shape and head shape.**

| Source | Degrees of Freedom | F-Value | *p*-Value |
|---|---|---|---|
| **Body Shape** | | | |
| Velocity environment | 1,2244 | 1.50 | 0.2201 |
| Centroid size (CS) | 1, 2244 | 46.72 | <0.0001 |
| Index | 11, 1334 | 29.25 | <0.0001 |
| Velocity environment*Index | 11, 1334 | 3.88 | <0.0001 |
| Centroid size*Index | 11, 1334 | 33.20 | <0.0001 |
| Velocity environment*CS*Index | 12, 1325 | 3.12 | 0.0002 |
| **Head Shape** | | | |
| Velocity environment | 1, 1723 | 0.02 | 0.8882 |
| Centroid size (CS) | 1, 1723 | 19.25 | <0.0001 |
| Index | 8, 1089 | 24.42 | <0.0001 |
| Velocity environment*Index | 8, 1089 | 10.73 | <0.0001 |
| Centroid size*Index | 8, 1089 | 24.26 | <0.0001 |
| Velocity environment*CS*Index | 9, 1075 | 7.14 | <0.0001 |

Centroid size represents ontogenetic variation.

at both ends of the ontogenetic spectrum (Fig 3). Smaller *T. areolatus* from the low-velocity environment had proportionally longer heads and deeper bodies and heads (dorsoventrally) compared to specimens from the high-velocity environment (Fig 4, upper); whereas, larger *T. areolatus* from the low-velocity environment had proportionally shorter heads compared to specimens from the high-velocity environment (Fig 4, lower).

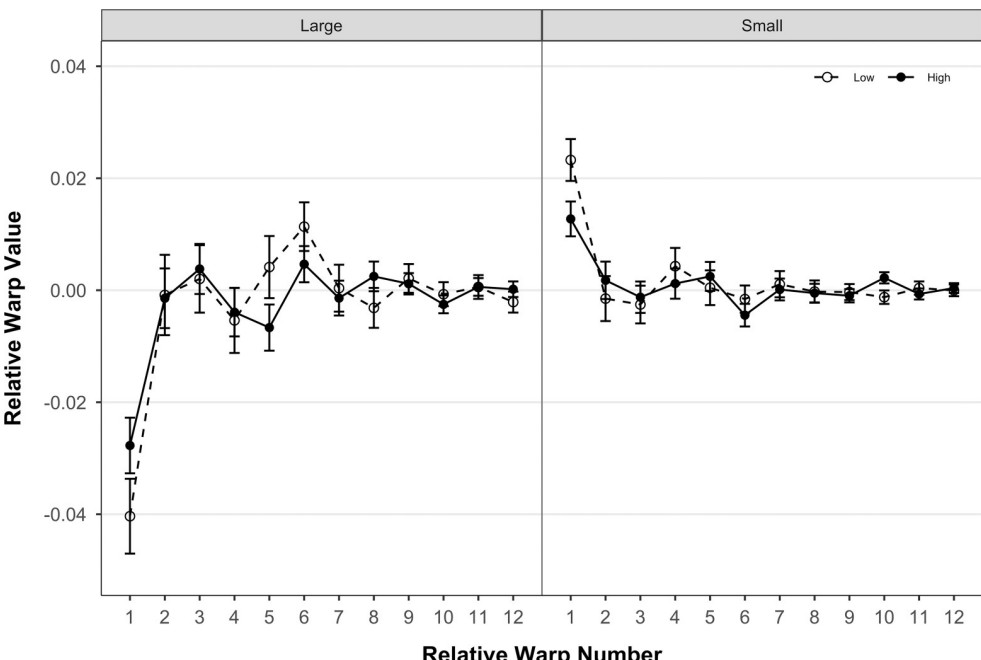

**Fig 3. Relative Warp least squares means for body.** Least squares means (error bars represent 95% confidence intervals of the mean) for each of 12 relative warps from the multivariate linear mixed model of body shape variation of *T. areolatus*. High-velocity environment is represented by closed circle symbols and solid connecting lines, and low-velocity environment is represented by open circle symbols and dashed connecting lines. Small and large ontogenetic stages are represented by centroid sizes of 35.00 and 150.00, respectively.

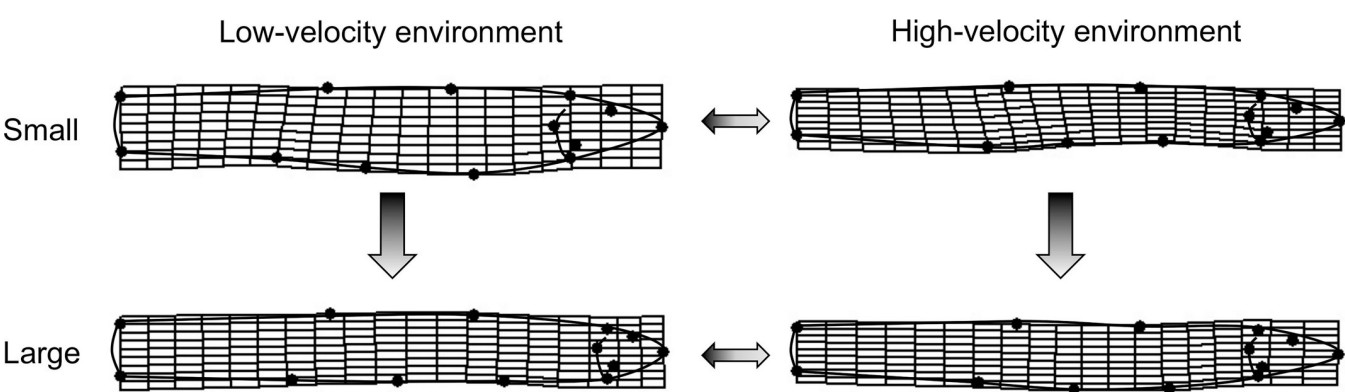

**Fig 4. Visualization of variation in body shape.** Thin plate spline deformations of divergence in body shape between high-velocity and low-velocity environments for large ontogenetic stage and small ontogenetic stage. Vertical arrows represent the direction of ontogenetic shape change from small to large. Horizontal arrows represent divergence between velocity environments.

Head shape of *T. areolatus* (dorsal view) differed significantly over ontogeny (centroid size), between velocity environments, and ontogeny and velocity environment exhibited a significant interaction (see the two-way and three-way interaction with the index variable in Table 2). Ontogenetic shape change in the head was associated with relative warps 1, 2, 3, 5, and 6. Shape differences in the head associated with velocity environment were evident on relative warps 1–7 (Fig 5). The direction of shape change of the head (p = 0.0024), but not the magnitude (p = 0.0886) differed significantly between velocity environments. The multivariate direction of the ontogenetic trajectory of head shape differed by 47.63˚ between velocity environments. This difference in direction of the ontogenetic trajectory of head shape between velocity environments represents a crossing norm of reaction in that relative warp scores for the large and small ends of the ontogenetic trajectory alternate in sign or magnitude between high and low-velocity environments (warps 1–6; Fig 5). Visually, the change in shape over ontogeny in the low-velocity environment involves a substantial reduction in head width and an especially marked reduction in relative eye size (Fig 6, left side). In contrast, the change in shape over ontogeny in the high-velocity environment involves an increase in head width and a slight elongation of the head anterior of the eye (Fig 6, right side). Head shape differs between high-velocity and low-velocity environments at both ends of the ontogenetic spectrum (Fig 5). Smaller *T. areolatus* from the high-velocity environment had narrower heads, and smaller eyes compared to small specimens from the low-velocity environment (Fig 6, upper); whereas, larger *T. areolatus* from the high-velocity environment had broader and longer heads, compared to specimens from the low-velocity environment (Fig 6, lower).

## Discussion

Velocity environments differentially shape the ontogenetic trajectory of shape in *T. areolatus*. Two patterns emerge from our data that inform how these two velocity environments differentially affect ontogenetic shape trajectories. First, in both head and body, differences in shape exist between environments at the smallest ontogenetic sizes. This pattern is similar to that observed in *B. rhabdophora* between high and low-predation environments—even the smallest individuals were divergent in shape between environments [51]. Development of shape is highly plastic in fishes [88], and induced effects from hatchery environments are especially well documented [89]. In cases where fish shape is induced by environment, shape typically does not differ at the earliest stages of ontogenetic development (before environmental effects

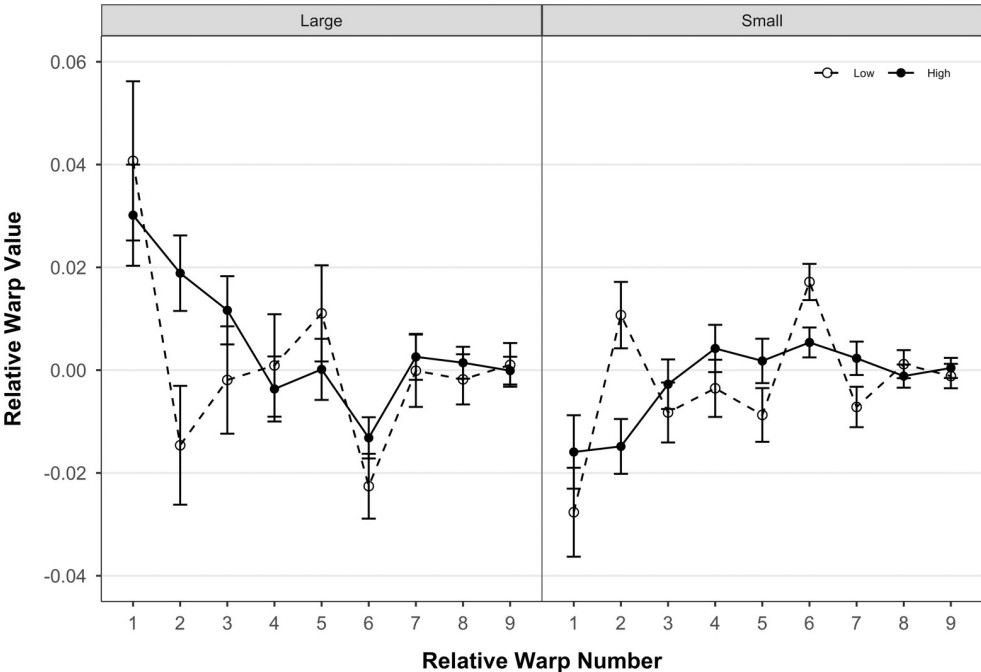

**Fig 5. Relative warp least squares means for head.** Least squares means (error bars represent 95% confidence intervals of the mean) for each of 9 relative warps from the multivariate linear mixed model of head shape variation of *T. areolatus*. The high-velocity environment is represented by closed circle symbols and solid connecting lines, and the low-velocity environment is represented by open circle symbols and dashed connecting lines. Small and large ontogenetic stages are represented by centroid sizes of 5.00 and 20.00, respectively.

are manifest), but rather diverges as a consequence of time spent in the divergent environments [88]. This is not consistent with our data. However, it could be that important shape divergence between environments occurs at the larval stage or the larval to juvenile transition [89] and is thus already manifest by the smallest stage at which we sampled.

The second important pattern is that the magnitude of overall shape change in both body and head is substantially lower in fish from the high-velocity environment compared to fish from the low-velocity environment. This suggests that shape at all stages may be more constrained in the high-velocity environment. Higher water velocities, along with other potentially selective effects, produce more convergent forms relative to environments with lower water velocities [90]. Where interactive effects have been identified in determining shape of fishes, one environmental effect is often more constraining than the other, thus creating conditions for significant interactions [17]. For example, Utah chub (*Gila atraria*) exhibit less variation in shape in response to diet in high predation environments, compared to low predation environments, suggesting that predation constrains fish shape more than diet [17]. Similarly, female *B. rhabdophora* from high predation and low predation environments converge in shape as adults, apparently in response to the demands of pregnancy on shape. Thus, pregnancy exerts a stronger effect on shape than does predation in females of this species [51]. In *T. areolatus*, high-velocity environments appear to constrain shape to a greater extent than low-velocity environments.

Multiple variables differ between velocity environments (i.e., stream zones [27]), and it is not clear from our data which of these variables are most important in determining shape variation over ontogeny. However, most studies that report differences in shape of stream fishes based on differences in environments focus on variation in stream velocity or surrogate

## Low-velocity environment    High-velocity environment

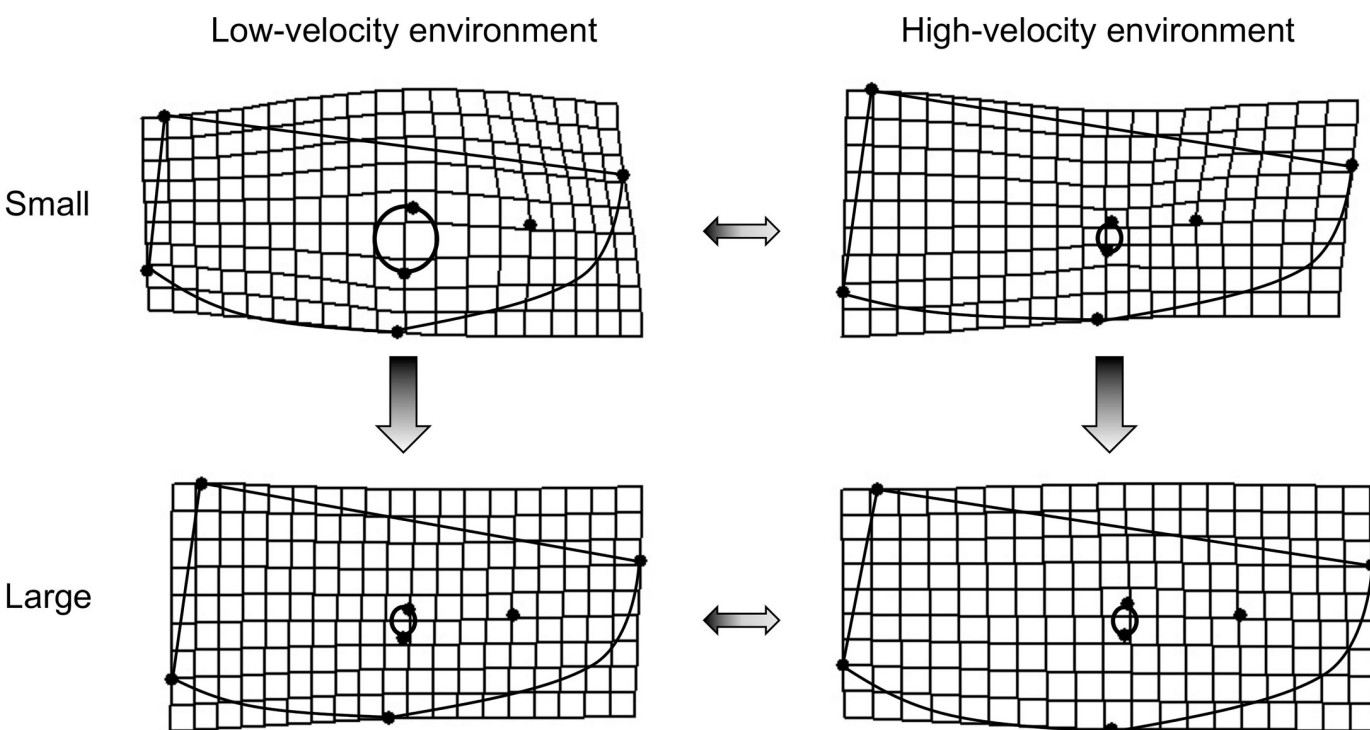

**Fig 6. Visualization of variation in head shape.** Thin plate spline deformations of divergence in head shape between high-velocity and low-velocity environments for large ontogenetic stage and small ontogenetic stage. Vertical arrows represent the direction of ontogenetic shape change from small to large. Horizontal arrows represent divergence between velocity environments.

variables such as stream gradient or flow rate [28–32]. Effects of water velocity are well documented [29–33]. Generally, narrower body and head shapes are favored in high-velocity environments which require steady swimming, whereas robust body and head shapes are favored in low-velocity environments which require unsteady swimming [28,91]. Our results were consistent with this pattern across the ontogenetic stages we sampled. *T. areolatus* sampled from high-velocity environments had narrower bodies and heads than *T. areolatus* sampled from low-velocity environments, and the general shape of body and head changed relatively little in the high-velocity environments. Overall, water velocity seems to be a likely constraint on variation in shape in the high-velocity environment, but water velocity appears to be relatively less of a constraint in its effect on shape in the low-velocity environment.

Generally, juvenile fish have larger heads (relative to body size) and narrower bodies, whereas adults have smaller heads (relative to body size) and more robust bodies [92–95]. Although proportional size of the head in *T. areolatus* changes over ontogeny consistent with the pattern observed in other studies, body shape does not become more robust with increasing size. This pattern of narrow body shape in adults may be associated with the benthic nature of *T. areolatus*, and changes in habitat use over ontogeny. Body shape of *T. areolatus* is convergent with body shape of other taxa that inhabit the benthic environment in high-velocity systems [90]. It may be that species that inhabit the hyporheic environment in high-velocity environments are constrained in shape such that more robust body forms may be selected against because of the small size of openings available in the substrate. A comparison of ontogenetic shape trajectories among these convergent species may lend support to this idea [90].

The presence of shape variation between velocity environments within the Andalién River is surprising, given the relatively small size of the river (48 km) and the apparent lack of

geographic or reproductive barriers to gene flow (i.e., no waterfalls or other flow barriers that would preclude movement between velocity environments). With no evidence of genetic isolation between velocity environments, the observed morphometric variation is unlikely to be based on underlying genetic differences. It is more likely an example of phenotypic plasticity in response to environmental conditions [24,25], but much of the induced difference must occur during the larval to juvenile transition at smaller sizes than we sampled. A common-garden experiment covering all sizes (larvae to adult) would be needed to directly address this question [96].

## Acknowledgments

We thank the undergraduate and graduate students at Brigham Young University and the Universidad de Concepción that helped collect, measure, and photograph the fish. We thank Jillian Campbell for landmarking the photos and preparing the data for analysis.

## Author Contributions

**Conceptualization:** Evelyn Habit, Mark C. Belk.

**Data curation:** Peter C. Searle, Mark C. Belk.

**Formal analysis:** Peter C. Searle, Mark C. Belk.

**Funding acquisition:** Evelyn Habit, Mark C. Belk.

**Investigation:** Evelyn Habit, Mark C. Belk.

**Methodology:** Evelyn Habit, Mark C. Belk.

**Project administration:** Evelyn Habit, Mark C. Belk.

**Resources:** Evelyn Habit.

**Supervision:** Mark C. Belk.

**Validation:** Mark C. Belk.

**Visualization:** Peter C. Searle, Margaret Mercer, Mark C. Belk.

**Writing – original draft:** Peter C. Searle, Margaret Mercer.

**Writing – review & editing:** Peter C. Searle, Margaret Mercer, Evelyn Habit, Mark C. Belk.

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
