## [Decision Letter · Decision Letter 0]

3 Dec 2020

PONE-D-20-33010

Ontogenetic shape trajectory of *Trichomycterus areolatus* varies in response to stream gradient

PLOS ONE

Dear Dr. Searle,

Thank you for submitting your manuscript to PLOS ONE. After careful consideration, we feel that it has merit but does not fully meet PLOS ONE’s publication criteria as it currently stands. Therefore, we invite you to submit a revised version of the manuscript that addresses the points raised during the review process.

I have now received two reviews. Both of them are as you can see fairly positive to the manuscript. However, both reviewers also pointed out that the manuscript needs substantial improvement, especially in the overall writing but there are also parts that need more explanation and justification. So, the success of the manuscript  critically depends on how you deal with the reviewer's comments in the review process.

In particular, you need to justify the selection procedure of relative warps in the statistical methods (e.g. lines 160-163, see the comments of reviewer 2). For example, it is not clear what morphological axis a significant interaction explains. There are also other parts in the description of the statistical methods that are confusing pointed out by both reviewers. Also for example lines 170-174 make little sense.

You also need to show more convincing evidence to whether the overall morphology pattern is driven by flow regimes or ontogeny alone, which is the major point of the article (see the comments of reviewer 1). This is especially important since currently it is not clear how you tease these two effects apart.

Both reviewers have also given a number of comments and suggestions and when you submit your revised manuscript you need to explicitly state, point by point in how you have dealt with the reviewers’ comments.

We look forward to receiving your revised manuscript.

Kind regards,

Peter Eklöv

Academic Editor

PLOS ONE

Journal Requirements:

3.In your Data Availability statement, you have not specified where the minimal data set underlying the results described in your manuscript can be found. PLOS defines a study's minimal data set as the underlying data used to reach the conclusions drawn in the manuscript and any additional data required to replicate the reported study findings in their entirety. All PLOS journals require that the minimal data set be made fully available. For more information about our data policy, please see http://journals.plos.org/plosone/s/data-availability.

Reviewers' comments:

Reviewer's Responses to Questions

**Comments to the Author**

1. Is the manuscript technically sound, and do the data support the conclusions?

Reviewer #1: Partly

Reviewer #2: Partly

2. Has the statistical analysis been performed appropriately and rigorously? 

Reviewer #1: Yes

Reviewer #2: I Don't Know

3. Have the authors made all data underlying the findings in their manuscript fully available?

Reviewer #1: Yes

Reviewer #2: Yes

4. Is the manuscript presented in an intelligible fashion and written in standard English?

Reviewer #1: Yes

Reviewer #2: Yes

5. Review Comments to the Author

Reviewer #1: Review: Ontogenetic shape trajectory of Trichomycterus areolatus varies in response to stream gradient

The study by Searle et al. uses morphometrics to investigate ontogenetic shape change in T. areolatus between two different flow environments. The authors find that ontogeny has the greatest effect on morphological shape change, followed by flow regime. The authors report that across environments, T. areolatus become more uniformly-shaped along the body and wider in the head with increasing size. Secondly, the authors find that between environments, high flow is associated with narrower bodies and heads laterally, but wider heads and larger eyes and shorter/rounder noses. The interpretation is that this is the acquisition of a more fusiform body shape in high flow regimes and is the result of phenotypic plasticity in response to environmental flow regime.

This is a nice study that is worthy of publication, but the writing and general interpretations need work. Be cautious with your interpretations – external landmarks are difficult to repeatedly place at homologous structures. Be sure to define your landmarks explicitly in the methods. Also, I wonder how much of the overall pattern is being driven by ontogeny alone. Could you consider comparing the adults alone to see if flow regime has an effect? Otherwise, it is tough to convince the readers that these populations differ with respect to flow regime.

Minor comments:

27 – among fishes?

49 –Do you mean developed or evolved?

50-52 – This is a tough transition. Consider the main topics in this paragraph and how they flow from one to another. Is this second paragraph about how fishes adapt to differing flow rates or is it about intraspecific adaptation?

53-56 – Do you need to define natural selection and phenotypic plasticity here? If so, consider your definition of natural selection.

91-94 – Can you provide more convincing reasoning behind grouping the transitional zone with the high flow? The gradient looks closer to the low flow. Generally, this seems problematic if you are trying to say something about how flow influences adaptive morphologies.

106-111 – Perhaps you can be more specific in terms of the breakdown of sizes? Bin them or tell the readers about the distribution? Reporting average size when you intentionally collected over a broad size range only tells the readers that the two groups have the same average size but nothing about the distribution of sizes.

111 – Can you provide details to the level that the method can be replicated? The resolution of photos makes a difference in the precision/accuracy of your morphometric analysis. Camera and lens make/model, etc.

122 – Is this the proper term for the rostral-most point on T. areolatus? Is the point at the tip of the premaxilla? Might try to be more specific here.

123 –Can you be more specific with Point 5 in terms of the importance of this point, why you placed it here and what it tells the reader?

135 – Can you be more specific about point 2? Is this the occipital crest? Remember that these points must be homologous for proper landmarking.

136-137 – It is unclear what point 6 means and how it is repeatedly placed. Is this the lateral-most boundary of the head at the level of the eye (i.e., the widest part of the head)? Perhaps you could say “lateral extent of the head indicated by the intersection of a line drawn perpendicular to points 1 & 2 that passes through the eye and the lateral aspect of the head along that line.” Or something more explicit.

162 – “12 rows (body) or 9 (head) rows” Consider being consistent about where parentheticals are placed in this sentence. It is helpful to the reader.

187-188 – Your explanation of Index methodology (above) is helpful in understanding how the interaction was developed. However, the way the topic sentence here is written is confusing/misleading. You state body shape differed significantly by flow and centroid size – but that is not what you report. I suggest you be explicit in how you open up the major finding of your results by stating exactly what was different, which is that body shape differs based on centroid size and between the three-way interaction of flow, centroid and index – this is different than saying that body shape differed sig by flow – which is what the reader takes away from this topic sentence, but is not necessarily true.

193-194 – Is ontogeny driving the larger patterns you are seeing, particularly with your index interaction? How can you tease apart the effect of flow within this? I’m afraid that the significance of your index and all interactions using that term are driven by the overarching effect of centroid size. Clarifying how you can separate out these effects would be helpful to readers.

198 – You report the magnitude of shape change is significantly different between flow regimes. Be sure to cite the statistical results supporting this comment.

210-211 – Same comment as line 187-188. The way you word this sentence is confusing and possibly misleading.

240 – The reporting of eye size is not obvious within Results. Is this true that adults have relatively larger eyes? I don’t see that in the warps and it doesn’t make sense with the allometry of eye size among other fishes. Eye size is strongly negatively allometric across ontogeny among fishes. Please check this reporting in your manuscript. Perhaps high flow adults have relatively larger eyes than low flow adults, but the way it is worded here suggests that eye size scales with positive allometry among high flow populations, and this should be clarified and expanded upon, if true.

241-242 – Be sure to include the proper statistics in reference to this comment. Additionally, does this mean that high flow fish retain juvenile features into adulthood? Could you comment on this from a heterochrony/paedomorphic standpoint? Perhaps this is how the environments are mediating the ontogenetic trajectory of these fish – by maintaining juvenile-like characteristics among the high flow populations.

250-251 – It looks like the dorsal view of the head for warp 2 shows a wider aspect of the head at the mouth. I do not disagree that this could still be adaptive for the high flow environment, but I would argue that it shows a more fusiform shape. You will need to be more explicit here and in the results on what the shape changes between environments really are.

259 – Check this report of adults with relatively smaller eyes with what you report previously.

265-267 – There is no citation for the comment on apparent lack of reproductive barriers to gene flow and isolation of populations. Is this personal observation?

269 – You might consider stating at the end of the sentence something along the lines that a common garden experiment would have to be performed to directly address this inference.

Figure 3 – It is difficult to see much difference among the warp line drawings. Consider a figure where they are made larger for the reader to see what is going on. Also, it is tough to interpret the data points on the graph. The data points are confined to as small section of the total space of the plot. Finally, low flow and high flow fish fell along perfectly flat, parallel lines? Perhaps zooming in could give the reader better resolution of the data points.

Reviewer #2: In this study the authors access body and head shape divergence of Trichomycterus areolatus between high and low flow river zones, comparing this change over a size range. They find both body and head shape is strong affected by size and, to a lesser extent, habitat zone. Interestingly, they show that the difference in head shape between high and low flow zones switches along size. Overall, I think the question and results are interesting and worth publishing. However, I think there is room for improving this manuscript. I find the introduction and discussion fairly narrow and does not set this study in a boarder context, especially for a journal like PLOS one that has a wide readership. In addition, the authors do not mention the many confounding factors that could influence body and head shape in this study design. Finally, the clarity of the methods could also be improved.

Specific comments:

Line 42: I would not use “swimming style” and use “swimming performance” instead. Swimming style suggests the fish change swimming form, such as anguilliform to carangiform.

Line 47: Remove “within species (i.e. intraspecific variation)”. There is not need to define intraspecific. It is a commonly used term.

Line 60: Again, why have “(i.e. juveniles)”? Just use juveniles in place of small individuals.

Line 74-82: Any information about the microhabitat use of Trichomycterus areolatus would be helpful here. Does it occur in habitats characterized by sand or wood structure; runs, riffles, or pools? For example, if this species only resides in pools habitat with high structure, any difference in river discharge may not influence this species since it would only reside in low flow.

Methods: This study compare individuals from sites with very different abiotic and biotic environments. This study design raises the concern that flow may not be the cause of any intraspecific shape differentiation between the zones, yet little is discussed on this topic.

Lines 92-94: Is a reason for combining the two zones into a high-flow grouping? Why not analyze all three separately? I would expect the high flow and low flow zone to have the greatest separation, with some overlap with the intermediate zone.

Lines 105-106: Why not use all the individuals?

Line 143: A reader not familiar with geometric morphometrics may need to know what non-shape variation is being removed: position, orientation, and scale.

Line 148-150: Why choose 12 and 9 axes? Is this based on a broken stick model, amount of variance explained, or some other method? This seem arbitrary and the figures in the manuscript only show 2 of the axes, seemly ignoring the rest that were included in the LMM.

Lines 160-163: This should be explained in more detail. I was confused by this section. I believe the authors created a vector (single column) that included all relative warp (RW) axes (12 or 9 rows) for each specimen so that: row 1 is RW1 for specimen 1, row 2 is RW2 for specimen 1, and so on. In this case, the dependent variable for the multivariate linear mixed model is a vector (single column) and not a matrix (multiple columns)?

It seems that any significant interaction between index variable and the environmental zone variable would show that at least one RW axis differs. However, we do not know which axis or the amount of variation explained by that axis. It is possible that only the 12th axis is significantly different which would explain a small proportion of the variation which and have very little meaningful impact on the overall shape. Typically, all 12 axes would be analyzed at once showing that the overall body shape differs between habitats. Many of the papers cited by the authors follow this method.

Lines 188-193: Where does this information come from? I am assuming Figure 1.

Lines 195-199: The figure shows a slight difference between low and high flow which is expected (higher flow having narrower bodies and heads) but the confidence intervals overlap a fair amount for relative warp 2. This applies to the magnitude of shape change too. Can you say these are significantly different? At least one of your body shape axes are different but that is likely relative warp 1 and maybe not relative warp 2.

Head shape shows clear patterns of separation. Interesting that larger fish had smaller heads in the body shape analysis but broader heads in the head shape analysis, being more compressed.

Lines 257-258: Please expand why this pattern may be associated with the benthic nature or habitat use changes.

6. PLOS authors have the option to publish the peer review history of their article (what does this mean?). If published, this will include your full peer review and any attached files.

Reviewer #1: No

Reviewer #2: No

---

## [Author Response · Author response to Decision Letter 0]

8 Feb 2021

I have now received two reviews. Both of them are as you can see fairly positive to the manuscript. However, both reviewers also pointed out that the manuscript needs substantial improvement, especially in the overall writing but there are also parts that need more explanation and justification. So, the success of the manuscript critically depends on how you deal with the reviewer's comments in the review process.

In particular, you need to justify the selection procedure of relative warps in the statistical methods (e.g. lines 160-163, see the comments of reviewer 2).

We have justified the use of a subset of relative warps as those that accounted for more than 1% of total shape variation, and we have cited papers that use this same selection criterion. See lines 180-182.

For example, it is not clear what morphological axis a significant interaction explains.

To be clear, our analysis is a multivariate mixed model, such that all selected relative warps are used simultaneously in the analysis. Thus, the interactions are based on the summed effect across multiple relative warps. To give an indication of how relative warps contributed to differences among groups, we included a new figure that plots least squares means for each relative warp for each size by flow environment combination. See lines 213-216 and Figures 3 and 5. We now explain in the results section how each relative warp is related to the overall shape change documented by the significant interaction terms. See lines 244-247 and 273-275. 

 There are also other parts in the description of the statistical methods that are confusing pointed out by both reviewers. Also for example lines 170-174 make little sense.

We have now revised this sentence to indicate that we are testing for shape change across all relative warps simultaneously. See lines 205-209. 

You also need to show more convincing evidence to whether the overall morphology pattern is driven by flow regimes or ontogeny alone, which is the major point of the article (see the comments of reviewer 1). This is especially important since currently it is not clear how you tease these two effects apart.

To be clear, the significant three-way interactions from our multivariate mixed model analysis (for both head and body) indicate that it is not just ontogeny or flow environment, but an interaction between the two. To make this clearer to the reader, we have added two new analyses to our paper. First, we have conducted a phenotypic change vector analysis that specifically tests for differences in magnitude and direction of multivariate shape change between flow environments over ontogeny. This analysis shows that in the body view ontogenetic shape change vectors differ significantly between low and high flow environments in magnitude, but not in direction. In the head view, ontogenetic shape change vectors differ significantly between low and high flow environments in direction, but not in magnitude. See lines 217-228 for an explanation of the methods of analysis and see lines 247-250 and 275-279 for the results of the analysis. The second analysis is a calculation of a divergence vector that allows us to visualize the maximum divergence due to flow environment at small and large sizes. We provide new thin-plate spline visualizations of these divergence patterns for comparison among groups. See figures 4 and 6 and lines 229-238 for an explanation of the methods of analysis and see lines 250-254 and 279-283 and for the results of the analysis. With the help of these two new analyses, we think the nature of this interesting interaction is made clear.

Both reviewers have also given a number of comments and suggestions and when you submit your revised manuscript you need to explicitly state, point by point in how you have dealt with the reviewers’ comments. 

In your Methods section, please provide additional information regarding the permits you obtained for the work. Please ensure you have included the full name of the authority that approved the field site access and, if no permits were required, a brief statement explaining why.

We have included a complete explanation of how collections were made and what guidelines we used. These collections were made in 2003 and 2004 and at that time in Chile this sampling was completed under the supervision of the Dirección de Investigación, Universidad de Concepción and the Undersecretariat of Fisheries with no formal permit being required. We have clearly stated the guidelines we used in lines 110-115. 

3.In your Data Availability statement, you have not specified where the minimal data set underlying the results described in your manuscript can be found. PLOS defines a study's minimal data set as the underlying data used to reach the conclusions drawn in the manuscript and any additional data required to replicate the reported study findings in their entirety. All PLOS journals require that the minimal data set be made fully available. For more information about our data policy, please see http://journals.plos.org/plosone/s/data-availability.

The data set used for analysis has been uploaded to Dryad https://datadryad.org/stash/share/lC_kykyvRxHgybaB10KANtwGOfBNtgf888BdqBmqDxw

(Private link). The public DOI will be provided upon formal acceptance of the manuscript. 

Comments to the Author

Review Comments to the Author

Reviewer #1: Review: Ontogenetic shape trajectory of Trichomycterus areolatus varies in response to stream gradient

The study by Searle et al. uses morphometrics to investigate ontogenetic shape change in T. areolatus between two different flow environments. The authors find that ontogeny has the greatest effect on morphological shape change, followed by flow regime. The authors report that across environments, T. areolatus become more uniformly-shaped along the body and wider in the head with increasing size. Secondly, the authors find that between environments, high flow is associated with narrower bodies and heads laterally, but wider heads and larger eyes and shorter/rounder noses. The interpretation is that this is the acquisition of a more fusiform body shape in high flow regimes and is the result of phenotypic plasticity in response to environmental flow regime.

This is a nice study that is worthy of publication, but the writing and general interpretations need work. Be cautious with your interpretations – external landmarks are difficult to repeatedly place at homologous structures. Be sure to define your landmarks explicitly in the methods. 

We have added clarifying text to tell how we reduced the amount of noise generated by inaccurate placement of landmarks. See lines 144-151. Briefly, we 1) carefully reviewed each photograph before landmarking, 2) had one researcher landmark specimens randomly without reference to collection location or body size, and 3) had two other researchers quality check the landmarks placed by one individual to confirm homologous placement of points. 

Also, I wonder how much of the overall pattern is being driven by ontogeny alone. Could you consider comparing the adults alone to see if flow regime has an effect? Otherwise, it is tough to convince the readers that these populations differ with respect to flow regime.

To be clear, the significant three-way interactions from our multivariate mixed model analysis (for both head and body) indicate that it is not just ontogeny or flow environment, but an interaction between the two. To make this clearer to the reader, we have added two new analyses to our paper. First, we have conducted a phenotypic change vector analysis that specifically tests for differences in magnitude and direction of multivariate shape change between flow environments over ontogeny. This analysis shows that in the body view ontogenetic shape change vectors differ significantly between low and high flow environments in magnitude, but not in direction. In the head view, ontogenetic shape change vectors differ significantly between low and high flow environments in direction, but not in magnitude. See lines 217-228 for an explanation of the methods of analysis and see lines 247-250 and 275-279 for the results of the analysis. The second analysis is a calculation of a divergence vector that allows us to visualize the maximum divergence due to flow environment at small and large sizes. We provide new thin-plate spline visualizations of these divergence patterns for comparison among groups. See figures 4 and 6 and lines 229-238 for an explanation of the methods of analysis and see lines 250-254 and 279-283 and for the results of the analysis. With the help of these two new analyses, we think the nature of this interesting interaction is made clear.

Minor comments:

27 – among fishes? 

Agreed. We changed “in fish” to “among fishes.” See line 27. 

 49 –Do you mean developed or evolved? 

The introduction has been significantly revised and we no longer use this phrase in the text. 

50-52 – This is a tough transition. Consider the main topics in this paragraph and how they flow from one to another. Is this second paragraph about how fishes adapt to differing flow rates or is it about intraspecific adaptation? 

We have rewritten much of the introduction to provide better flow among paragraphs. See lines 46-56. 

53-56 – Do you need to define natural selection and phenotypic plasticity here? If so, consider your definition of natural selection.

For readers of Plos, we agree that the definitions for natural selection and phenotypic plasticity are not needed. They have been removed from the manuscript. 

91-94 – Can you provide more convincing reasoning behind grouping the transitional zone with the high flow? The gradient looks closer to the low flow. Generally, this seems problematic if you are trying to say something about how flow influences adaptive morphologies.

We have revised and added to the text to clarify our division of high and low flow environments. We state that because densities are low in the rhithron zone we had to combine samples with the transitional zone, and that there are other covarying variables other than flow that might affect shape of T. areolatus. These concepts are made clear in lines 103-108 and 116-123. 

106-111 – Perhaps you can be more specific in terms of the breakdown of sizes? Bin them or tell the readers about the distribution? Reporting average size when you intentionally collected over a broad size range only tells the readers that the two groups have the same average size but nothing about the distribution of sizes.

Thank you for this suggestion. We added in size ranges for each of the locations for both the head and lateral analyses. See lines 131-137. 

111 – Can you provide details to the level that the method can be replicated? The resolution of photos makes a difference in the precision/accuracy of your morphometric analysis. Camera and lens make/model, etc. 

We added in additional information about which camera and lens make/model we used. See lines 127-128

122 – Is this the proper term for the rostral-most point on T. areolatus? Is the point at the tip of the premaxilla? Might try to be more specific here. 

We changed the wording from “anterior tip of snout” to “rostral-most point.” See line 152. 

123 –Can you be more specific with Point 5 in terms of the importance of this point, why you placed it here and what it tells the reader? 

We use this landmark in conjunction with landmark 4 to estimate depth of the head. We do not feel it is necessary to provide additional information in the text about what this landmark captures. If we explain the purpose of this one landmark, we feel it would be necessary to provide descriptions for every single landmarks. See lines 154-155. 

135 – Can you be more specific about point 2? Is this the occipital crest? Remember that these points must be homologous for proper landmarking. 

This point represents the midline of the head at the back of the head. It is positioned on the midline as projected from point number 7. We modified the description of point two in the text. See lines ###

136-137 – It is unclear what point 6 means and how it is repeatedly placed. Is this the lateral-most boundary of the head at the level of the eye (i.e., the widest part of the head)? Perhaps you could say “lateral extent of the head indicated by the intersection of a line drawn perpendicular to points 1 & 2 that passes through the eye and the lateral aspect of the head along that line.” Or something more explicit. 

This point is used to measure the width of the head at the eye. We modified the description of point 6 in the text. See lines 166-167. 

162 – “12 rows (body) or 9 (head) rows” Consider being consistent about where parentheticals are placed in this sentence. It is helpful to the reader.

We updated the sentence to “12 rows (body) or 9 rows (head) to keep the parentheticals consistent. Thank you for catching this inconsistency. See line 196. 

187-188 – Your explanation of Index methodology (above) is helpful in understanding how the interaction was developed. However, the way the topic sentence here is written is confusing/misleading. You state body shape differed significantly by flow and centroid size – but that is not what you report. I suggest you be explicit in how you open up the major finding of your results by stating exactly what was different, which is that body shape differs based on centroid size and between the three-way interaction of flow, centroid and index – this is different than saying that body shape differed sig by flow – which is what the reader takes away from this topic sentence, but is not necessarily true. 

We have revised this sentence to more clearly state the statistical result of significance for all two-way and the three-way interactions. See lines 241-244. 

193-194 – Is ontogeny driving the larger patterns you are seeing, particularly with your index interaction? How can you tease apart the effect of flow within this? I’m afraid that the significance of your index and all interactions using that term are driven by the overarching effect of centroid size. Clarifying how you can separate out these effects would be helpful to readers. 

To be clear, the significant three-way interactions from our multivariate mixed model analysis (for both head and body) indicate that it is not just ontogeny or flow environment, but an interaction between the two. To make this clearer to the reader, we have added two new analyses to our paper. First, we have conducted a phenotypic change vector analysis that specifically tests for differences in magnitude and direction of multivariate shape change between flow environments over ontogeny. This analysis shows that in the body view ontogenetic shape change vectors differ significantly between low and high flow environments in magnitude, but not in direction. In the head view, ontogenetic shape change vectors differ significantly between low and high flow environments in direction, but not in magnitude. See lines 217-228 for an explanation of the methods of analysis and see lines 247-250 and 275-279 for the results of the analysis. The second analysis is a calculation of a divergence vector that allows us to visualize the maximum divergence due to flow environment at small and large sizes. We provide new thin-plate spline visualizations of these divergence patterns for comparison among groups. See figures 4 and 6 and lines 229-238 for an explanation of the methods of analysis and see lines 250-254 and 279-283 and for the results of the analysis. With the help of these two new analyses, we think the nature of this interesting interaction is made clear.

198 – You report the magnitude of shape change is significantly different between flow regimes. Be sure to cite the statistical results supporting this comment.

We have included a phenotypic change vector analysis to test for statistical significance in our three-way interaction. See lines 247-250.

210-211 – Same comment as line 187-188. The way you word this sentence is confusing and possibly misleading.

We have revised this sentence to more clearly state the statistical result of significance for all two-way and the three-way interactions. See lines 270-273

240 – The reporting of eye size is not obvious within Results. Is this true that adults have relatively larger eyes? I don’t see that in the warps and it doesn’t make sense with the allometry of eye size among other fishes. Eye size is strongly negatively allometric across ontogeny among fishes. Please check this reporting in your manuscript. Perhaps high flow adults have relatively larger eyes than low flow adults, but the way it is worded here suggests that eye size scales with positive allometry among high flow populations, and this should be clarified and expanded upon, if true. 

Saying that adults had larger eyes was a mistake. Our updated figures represent the effect of ontogeny in each flow environment across all relative warps. We changed the results and discussion so that it is clear that change in eye size is part of the interaction between ontogeny and flow environment. See lines 325-330.

241-242 – Be sure to include the proper statistics in reference to this comment. Additionally, does this mean that high flow fish retain juvenile features into adulthood? Could you comment on this from a heterochrony/paedomorphic standpoint? Perhaps this is how the environments are mediating the ontogenetic trajectory of these fish – by maintaining juvenile-like characteristics among the high flow populations.

We have included a phenotypic change vector analysis to test for statistical significance in our three-way interaction. See lines 275-279. We have commented on potential reasons for differences in shape between flow environments over ontogeny. Since this is a descriptive study and we are only commenting on the shape of these organism, we do not feel comfortable calling this a paedomorphic change. Identification of paedomorphism/heterochrony requires a more detailed sampling of patterns of shape variation associated with reproductive maturity. 

250-251 – It looks like the dorsal view of the head for warp 2 shows a wider aspect of the head at the mouth. I do not disagree that this could still be adaptive for the high flow environment, but I would argue that it shows a more fusiform shape. You will need to be more explicit here and in the results on what the shape changes between environments really are. 

We have completely revised how we present our results. We have run two additional analyses outlined above and have been careful to integrate our explanation of body and head shape. See lines 250-254 and 279-283. 

259 – Check this report of adults with relatively smaller eyes with what you report previously

Thanks for this comment. Adults do in fact have relatively smaller eyes compared to juveniles. 

265-267 – There is no citation for the comment on apparent lack of reproductive barriers to gene flow and isolation of populations. Is this personal observation?

We have modified the text to indicate that there are no obvious barriers. See lines 341-344.

269 – You might consider stating at the end of the sentence something along the lines that a common garden experiment would have to be performed to directly address this inference.

We added an additional paragraph that discusses how a carefully designed, common-garden experiment would be useful in further addressing our inference. See lines 331-341 and lines 346-348

Figure 3 – It is difficult to see much difference among the warp line drawings. Consider a figure where they are made larger for the reader to see what is going on. Also, it is tough to interpret the data points on the graph. The data points are confined to as small section of the total space of the plot. Finally, low flow and high flow fish fell along perfectly flat, parallel lines? Perhaps zooming in could give the reader better resolution of the data points. 

We generated new thin plate spline deformations from our new analyses. We made the thin plate spline images much larger so that the reader can easily see the morphometric changes that occur between environments and across ontogeny. See figures 4 and 6.

Reviewer #2: In this study the authors access body and head shape divergence of Trichomycterus areolatus between high and low flow river zones, comparing this change over a size range. They find both body and head shape is strong affected by size and, to a lesser extent, habitat zone. Interestingly, they show that the difference in head shape between high and low flow zones switches along size. Overall, I think the question and results are interesting and worth publishing. However, I think there is room for improving this manuscript. I find the introduction and discussion fairly narrow and does not set this study in a boarder context, especially for a journal like PLOS one that has a wide readership. 

We have made major revisions to the introduction and discussion to broaden the scope of the manuscript. We have illustrated how morphometric traits can be selected for by the environment across many taxa, in addition to fish. See lines 38-51 and 298-307. 

In addition, the authors do not mention the many confounding factors that could influence body and head shape in this study design. 

We clearly state that other confounding factors could be present in the environment and could affect shape changes seen in T. areolatus. We have included explanations both in the methods and the discussion in our manuscript. See lines 104-108 and 331-340. 

Finally, the clarity of the methods could also be improved.

To be clear, the significant three-way interactions from our multivariate mixed model analysis (for both head and body) indicate that it is not just ontogeny or flow environment, but an interaction between the two. To make this clearer to the reader, we have added two new analyses to our paper. First, we have conducted a phenotypic change vector analysis that specifically tests for differences in magnitude and direction of multivariate shape change between flow environments over ontogeny. This analysis shows that in the body view ontogenetic shape change vectors differ significantly between low and high flow environments in magnitude, but not in direction. In the head view, ontogenetic shape change vectors differ significantly between low and high flow environments in direction, but not in magnitude. See lines 217-228 for an explanation of the methods of analysis and see lines 247-250 and 275-279 for the results of the analysis. The second analysis is a calculation of a divergence vector that allows us to visualize the maximum divergence due to flow environment at small and large sizes. We provide new thin-plate spline visualizations of these divergence patterns for comparison among groups. See figures 4 and 6 and lines 229-238 for an explanation of the methods of analysis and see lines 250-254 and 279-283 and for the results of the analysis. With the help of these two new analyses, we think the nature of this interesting interaction is made clear.

Specific comments:

Line 42: I would not use “swimming style” and use “swimming performance” instead. Swimming style suggests the fish change swimming form, such as anguilliform to carangiform.

The introduction has been significantly revised and we no longer use this phrase in the text. 

Line 47: Remove “within species (i.e. intraspecific variation)”. There is not need to define intraspecific. It is a commonly used term. 

The introduction has been significantly revised and we no longer use this phrase in the text. 

Line 60: Again, why have “(i.e. juveniles)”? Just use juveniles in place of small individuals. 

We removed “(i.e. juveniles)” as suggested. 

Line 74-82: Any information about the microhabitat use of Trichomycterus areolatus would be helpful here. Does it occur in habitats characterized by sand or wood structure; runs, riffles, or pools? For example, if this species only resides in pools habitat with high structure, any difference in river discharge may not influence this species since it would only reside in low flow. 

Thank you for this suggestion. Trichomycterus areolatus does in fact occur in runs and riffles where there is rapid water velocity. We have added additional information on the microhabitat use of T. areolatus in the text. See lines 85-86. 

Methods: This study compare individuals from sites with very different abiotic and biotic environments. This study design raises the concern that flow may not be the cause of any intraspecific shape differentiation between the zones, yet little is discussed on this topic. 

Our revised manuscript clearly states that other confounding factors could be present in the environment and could affect shape changes seen in T. areolatus. We have included explanations both in the methods and the discussion in our manuscript. See lines 104-108 and 331-340.

Lines 92-94: Is a reason for combining the two zones into a high-flow grouping? Why not analyze all three separately? I would expect the high flow and low flow zone to have the greatest separation, with some overlap with the intermediate zone. 

We have revised and added to the text to clarify our division of high and low flow environments. We state that because densities are low in the rhithron zone we had to combine samples with the transitional zone, and that there are other covarying variables other than flow that might affect shape of T. areolatus. These concepts are made clear in lines 103-108 and 116-123. 

Lines 105-106: Why not use all the individuals? 

Thank you for asking for clarification on this sentence. “Randomly selected” does not provide an accurate description of how we reached our final number of specimens collected. In fact, we used all of the specimens that were available to us unless they were preserved poorly. We have made updates in the text that clearly indicates how we selected the individuals that we used. See lines 128-129. 

Line 143: A reader not familiar with geometric morphometrics may need to know what non-shape variation is being removed: position, orientation, and scale.

We added in that non-shape variation includes position, orientation and scale in the text for readers unfamiliar with geometric morphometrics. See lines 174-175. 

Line 148-150: Why choose 12 and 9 axes? Is this based on a broken stick model, amount of variance explained, or some other method? This seem arbitrary and the figures in the manuscript only show 2 of the axes, seemly ignoring the rest that were included in the LMM. 

We have justified the use of a subset of relative warps as those that accounted for more than 1% of total shape variation, and we have cited papers that use this same selection criterion. See lines 180-182.We also included an analysis that generates thin-plate spline deformations across all relative warps. See figures 4 and 6 and lines 229-238. 

Lines 160-163: This should be explained in more detail. I was confused by this section. I believe the authors created a vector (single column) that included all relative warp (RW) axes (12 or 9 rows) for each specimen so that: row 1 is RW1 for specimen 1, row 2 is RW2 for specimen 1, and so on. In this case, the dependent variable for the multivariate linear mixed model is a vector (single column) and not a matrix (multiple columns)? 

We added additional detail to the explanation of vectorization to specifically indicate that the response variable is transposed from a row in the matrix to a vector column of responses. See lines 197-198. We believe this makes the vectorization process clearer and we appreciate the suggestion.

Lines 188-193: Where does this information come from? I am assuming Figure 1.

Since we used two new analyses and generated new figures, the results have been significantly modified. We now provide specific statistical tests and visualizations of the effects of flow environment on ontogenetic shape change. See lines 241-254 and See figures 3 and 4. 

Lines 195-199: The figure shows a slight difference between low and high flow which is expected (higher flow having narrower bodies and heads) but the confidence intervals overlap a fair amount for relative warp 2. This applies to the magnitude of shape change too. Can you say these are significantly different? At least one of your body shape axes are different but that is likely relative warp 1 and maybe not relative warp 2. 

To be clear, our analysis is a multivariate mixed model, such that all selected relative warps are used simultaneously in the analysis. Thus, the interactions are based on the summed effect across multiple relative warps. To give an indication of how relative warps contributed to differences among groups, we included a new figure that plots least squares means for each relative warp for each size by flow environment combination. See lines 213-216 and Figures 3 and 5. We now explain in the results section how each relative warp is related to the overall shape change documented by the significant interaction terms. See lines 244-247 and 273-275. 

Head shape shows clear patterns of separation. Interesting that larger fish had smaller heads in the body shape analysis but broader heads in the head shape analysis, being more compressed. 

We have included new thin plate spline deformations to clearly illustrate the shape changes we see across all relative warps. See figures 4 and 6. These figures show that adults do in fact have shorter heads relative to the body that are broader. See lines 250-254 and 281-283.

Lines 257-258: Please expand why this pattern may be associated with the benthic nature or habitat use changes. 

We have significantly modified the discussion. We added several sentences expanding how this pattern may be associated with the benthic nature of T. areolatus. See lines 308-324.

---

## [Decision Letter · Decision Letter 1]

23 Feb 2021

PONE-D-20-33010R1

Ontogenetic shape trajectory of *Trichomycterus areolatus* varies in response to stream gradient

PLOS ONE

Dear Dr. Searle,

Thank you for submitting your manuscript to PLOS ONE. After careful consideration, we feel that it has merit but does not fully meet PLOS ONE’s publication criteria as it currently stands. Therefore, we invite you to submit a revised version of the manuscript that addresses the points raised during the review process.

I have now received a second review of your manuscript and I agree with the reviewer that your manuscript has improved but there are still major alterations needed to reach a publishable state. These are outlined by the very detailed attached review with  essentially three major points: (1) you still need to clarify much of the writing. The attached review has supplied detailed suggestions in how this can be achieved (2) you need to supply further interpretation on how environment mediates ontogenetic trajectory and (3) to reconcile the stream gradient issue. The two latter points are especially important since these are related to the major take-home message.  

We look forward to receiving your revised manuscript.

Kind regards,

Peter Eklöv

Academic Editor

PLOS ONE

Reviewers' comments:

Reviewer's Responses to Questions

**Comments to the Author**

1. If the authors have adequately addressed your comments raised in a previous round of review and you feel that this manuscript is now acceptable for publication, you may indicate that here to bypass the “Comments to the Author” section, enter your conflict of interest statement in the “Confidential to Editor” section, and submit your "Accept" recommendation.

Reviewer #1: (No Response)

2. Is the manuscript technically sound, and do the data support the conclusions?

Reviewer #1: Yes

3. Has the statistical analysis been performed appropriately and rigorously? 

Reviewer #1: Yes

4. Have the authors made all data underlying the findings in their manuscript fully available?

Reviewer #1: Yes

5. Is the manuscript presented in an intelligible fashion and written in standard English?

Reviewer #1: Yes

6. Review Comments to the Author

Reviewer #1: (No Response)

7. PLOS authors have the option to publish the peer review history of their article (what does this mean?). If published, this will include your full peer review and any attached files.

Reviewer #1: No

---

## [Author Response · Author response to Decision Letter 1]

9 Apr 2021

Review: Ontogenetic shape trajectory of Trichomycterus areolatus varies in response to stream gradient

Re-Review:

The revision by Searle et al. is improved from the first submission, but the writing needs clarification throughout the manuscript. I have detailed line-by-line locations in the manuscript where the authors can consider clarifying, bolstering or specifying their arguments.

I still have a major issue with the take-home message being that stream gradient is the major feature driving the ontogenetic trajectories between the two groups. This is because your metric for quantifying stream gradient is the degree of river slope, which is 0.14 in the low-flow and 0.4/0.2 in the “high flow.” I understand that you grouped transitional flow with high so that you could have sufficient datasets for comparison, but I am not convinced this was the right way to go. Why combine transition flow data with high flow when the river slope in the transition is 0.2 degrees? This is closer to 0.14 of the low flow. Can you provide other data for flow in the low vs. high regimes, such as flow rate or flow volume? Your argument is that stream gradient is driving different ontogenetic trajectories in this species of fish. This is a super cool take-home message, so it needs to be clear that this is indeed the case.

We have thought through this argument and have gone back to a previous paper that detailed the variation among stream zones and our sampling locations. In the end, we agree with the reviewer that using gradient as our selective effect may not be the right way to present these differences. Our original intent was to represent variation in stream velocity, and we had used gradient as a surrogate, but further research suggested that gradient does not represent stream velocity very well in this system. Rather, we have now gone back and detailed the relevant differences between stream zones (see addition of Table 1) and have characterized the main difference as stream velocity itself. As it turns out, in this system, stream velocity is most different between the transitional and potamal zones, and stream velocity is the variable that many studies have used to characterize selective differences among stream environments. 

To answer the question, we have rewritten major sections of the introduction and methods to make clear that stream velocity is the main variable we use to characterize differences in environment. We acknowledge that in this sort of study there are likely multiple, covarying variables that may contribute to differences in shape between environments. We cannot differentiate among these variables in our sampling design (because they do covary), so instead we have characterized the environmental differences as “water velocity environments” to include the general environment that is created by high-water velocity. This same reasoning strategy has been used successfully in other studies that rely on “natural experiments” or sampling of variation across multiple environments. Studies on differences in “predator environments” or “competitor environments” acknowledge that although predators (for example) are likely to have a dominant effect in trait selection, other covarying variables may contribute to or oppose such selection. We rely on the same argument in this paper – although water velocity has been shown to influence shape of fishes, other variables may exert some selective influence as well.

We make these arguments in paragraph 3 in the introduction, we are more specific about them in paragraph 1 of the methods (study site characterization), and we discuss the implications in paragraph 4 of the discussion. As you can see, we have made major revisions to the paper to focus on water velocity environments as the relevant form of environmental variation. We appreciate the reviewer’s insights on this point, and we believe that the paper now makes a strong argument for differences in shape between selective environments characterized as high-velocity environments or low-velocity environments. 

Somewhat related, we have also addressed the issue by running the overall analysis with and without the few individuals from the rithron. We have demonstrated that the results we show are not dependent on the samples from the rithron. The statistical results and the estimates of the lsmeans are nearly identical, so we report the analysis from the full dataset (see the second paragraph in the methods). 

The second major issue is in the interpretation or importance/novelty of the findings that ontogenetic trajectory is mediated by environment. The statistics indicate that this is the case (through the interaction), but there is no further clarity or understanding of what this means in terms of HOW ontogeny is redirected in a low- vs. high-flow environment. What happens to the shape of these fish during development? I understand that high-flow are more fusiform with longer heads, but you don’t need ontogenetic data to show this. What is new from this study is that you can detail what specific shape changes occur throughout ontogeny between the two environments. You might consider bolstering this argument.

To answer this concern, we have now done a better job of describing the specific ontogenetic trajectories that occur in each of the water velocity environments. We describe these trajectories as differing in both beginning shape (at small sizes) and ending shape (at large sizes). We then describe how the trajectories differ in multivariate magnitude and direction of shape change. Whereas, before we just gave the statistical outcome of the trajectory tests, we now give a description of each trajectory. We describe how shape change over ontogeny in the high-velocity environment is constrained compared to that in the low-velocity environment. These descriptions add substantially to the results, Furthermore, we now discuss these two characteristics of ontogenetic shape change in separate paragraphs in the discussion. Paragraphs 2 and 3 in the discussion compare, contrast, and explore the patterns of shape change we observe. We compare these patterns to other similar studies to put them in context. 

I totally understand that it is deflating to receive a less-than-laudatory review. Your paper is getting there, but it needs (1) clarity in the writing, (2) further interpretation on how environment mediates ontogenetic trajectory (this is the novelty of your dataset) and (3) to reconcile the stream gradient issue (especially if this is the take-home message, i.e., in the title of your paper).

We appreciate the review, and it has led us to strive to clarify our writing to make the strength of this study apparent to readers. We have changed the focus of the environmental influence argument, and we have changed the title accordingly. We have further described and discussed how ontogenetic trajectories change between water velocity environments.

General comment not touched on below:

- Wider heads and larger eyes in high flow could also have to do with prey type in high- flow environments. Prey type is a potential source of selection that could produce similar results.

The reorganization of the discussion no longer includes the elements mentioned above. However, we have added the idea that prey type is a potential source of selection to the introduction. See lines 58-60. 

Line-specific comments:

27 – Sentence remains confusing (and it is important that this sentence is clear). Are you saying that head shape and body shape vary with respect to one another? This is kind of how this sentence reads.

Body and head shape can vary with respect to one another, but that is not the message we are trying to convey. We modified the sentence to indicate that body and head shape both vary in response to the water velocity environment over ontogeny. See lines 27-28

46 – This paragraph is on intraspecific variation in morphology, while the previous is on interspecific variation? I think you could clarify this for the reader. Also, you might include what the selective pressures were to cause these intraspecific morphological divergences in each example you provide.

We now use the words interspecific and intraspecific variation to make the main point of each paragraph clear to the reader. See lines 40-41 and 49-50. We added in additional information about the various habitats and selective pressures each study analyzed. See lines 50-54. 

50 – Finally, for this paragraph, you might be careful with your wording in the last sentence. By suggesting that phenotypic variation can result from either natural selection or phenotypic plasticity, you are implying that plasticity is not under the control of natural selection, which I would argue is not true. Perhaps what you mean to say is that variation can have a genetic or plastic component? If this is true, you should clarify this language.

We modified the sentence as suggested to clarify the meaning. See lines 54-55. 

52 – Your paragraph on ontogenetic shape change does not do justice to the vast literature on ontogenetic allometry among fishes, let alone vertebrates. From On Growth and Form (1917) to present day there are thousands of studies on the ontogeny of form. It would behoove you to give this subject more treatment both in the introduction and discussion of your paper.

We modified the paragraph in the introduction to include references to the many studies on ontogeny of form. See lines 70-73. 

74 – “select for” or “result in/produce?” Select for implies a specific understanding of adaptive significance behind phenotypic change. In this case you are saying that environment would select for a particular ontogenetic strategy (i.e., fast growth rate or something). I think what you are trying to say is that environment can mediate ontogenetic trajectory to produce a different adult morphology from a similar juvenile starting morphology?

We now use different wording to clarify that environments influence ontogenetic trajectories. This is the basis of the interaction. See lines 93-95. 

74 – 78 – It is unclear what is happening to male/female morphology between high and low predation environments. Are males the same throughout development or do they start different and undergo parallel shape change? Females start different but end up the same (converge)? This need clarification.

We clarified this sentence so that it states that the shape differences between predation environment in juvenile males is maintained in adult males; whereas, the shape differences between juvenile females is less pronounced in mature females. Females converge on a common shape as a consequence of pregnancy. See lines 84-90. 

78 – 80 – Are you saying here that in this stickleback study the three environments produced three different morphologies? Or that marine differed from fresh? Please clarify. Also, is it that these studies did not look at shape change throughout development and that your study is doing this? It would be good for you to clarify exactly what has been done and where your study fits in.

In this study, three different ontogenetic trajectories were observed. Although two marine habitats were used, they differed from each other. See lines 90-92. Both of these studies did look at shape change throughout development. However, both of these studies did not specifically quantify the interaction between water velocity environment across ontogenetic trajectories in fishes. See lines 90-93. 

80 – 82 – Consider deleting “in different environments over ontogeny” – this part of the sentence is repetitive.

We removed the phrase as suggested. See lines 93-95. 

107 – Are you saying here that you grouped the environments into high-flow and low-flow and acknowledge that there are other features of these environments that might be causing the observed shape change that you quantify (other than flow)? You ultimately suggest (in the title) that shape change is a product of the stream gradient (which definitely implies flow), and not the confounding variables that also vary with environment.

I remain unconvinced of your reasoning to group the transitional zone with the high-flow. Your whole argument is in how flow speed shapes morphology, but the only values you present on flow speed are the slopes of these different zones (0.4, 0.2 and 0.14 degrees). You group the

0.4 and 0.2 together (lines 120-123) and compare this to the 0.14 and find difference in morphology and this is great, but I don’t know what to make of those differences in a functional way as a result of the “two different environments.” Can you provide a metric of flow velocity among the three sites, or flow volume (cubic feet per second) to better distinguish low- vs.

high-flow?

We have done as suggested here and as explained above this has resulted in substantial revisions to our arguments. See the more complete explanation above. 

131 & 134 – Were these two totally different data sets for body shape and head shape? Or did you use mostly the same specimens, just one less for the head shape and 10 more for body shape? Also, were individuals tracked to compare head and body shape of the same individual? This would be important in the statistical analysis (i.e., repeated measures).

We used mostly the same specimens for both body and head shape. We have added an explanation as to how some individuals may be suitable for the head view but not the body and vice versa. See lines 222-223. We did not track individuals to compare joint shape change between body and head views, but the data are broadly overlapping. Each of the statistical analyses (head and body) were run independently, thus there is not statistical dependence here. The repeated measures analogy is based on multiple shape variables used in the analysis at one time, but we did not mix head and body relative warps in analyses. In the figures that show the shape variation associated with divergence vectors, one can see that patterns of shape change are consistent between body and head. For example, short heads relative to bodies in the low-velocity environment (body view) is the same as shorter heads overall in the head view in the same environment. 

298 – I think I know what you are trying to say with this opening sentence, but it is awkward. It reads as if ontogeny is a selective pressure. Same with life history and environment. I believe that environment can provide selection, but I’m not sure that ontogeny and life histories are selective pressures so much as they are traits that can adapt as a result of selection. You should consider opening this section with a strong statement of your major result.

We have revised this confusing sentence. We now emphasize that environment is the selective force we are interested in analyzing across ontogenetic trajectories. See lines 347-348.

306 – You might consider being more specific here with what your results can say about how environment shapes ontogeny. You do not discern between genetic basis (that these are distinct populations that have evolved different developmental programs) and plasticity (that these populations are responding to their environment). It would take another study to tease these effects apart, but it could be worth acknowledging or specifying here what your data are capable of saying.

We have now specifically described how ontogenetic shape trajectories vary between water velocity environments. We have detailed this above. In addition, we have been clearer that we cannot tell whether these changes have a genetic or plastic basis, or both, and that it would take a common-garden environment experiment to sort this out (as well as the specific effect of each environmental variable associated with water velocity. 

308 – What is your metric for fusiform shape? How was this calculated? The only other mention of “fusiform” is in the introduction section so it is hard for the reader to determine how you calculated this variable.

We used fusiform as a general descriptor to indicate narrower relative to more robust bodies. We have revised this sentence to use these more qualitative terms. See lines 388-390. 

312 – Consider a new paragraph when transitioning to juveniles. This is a massive paragraph otherwise.

The discussion paragraphs have been substantially revised and restructured. This paragraph has been split up accordingly. See lines 397-407. 

321 – Is “endorheic” the correct term here (closed drainage basin)? Does your study river terminate into a lake/basin? For those unfamiliar this might be worth clarifying. Also, you are talking about high flow environments in the preceding sentence, so a clearer transition is needed. 327 – Negative allometry in eye size (relatively large eyes in juveniles) is observed in most ontogenetic studies of fish and may be a developmental constraint more than a product of adaptation.

Endorheic is not the correct term. Thank you for catching this, we are embarrassed by this mistake. We now use the term hyporheic which was our original meaning. See lines 404-406. We modified the sentence so that the transition from high-velocity environment to benthic environments is clearer. See lines 402-406. The reorganization of the discussion no longer includes the elements mentioned above. However, we agree that negative allometry in eye size is likely a developmental constraint. 

332 – 335 – Please clarify what you are saying here. Is the high flow environment punctuated by open basins of little-to-no flow? Are you saying that the benthic substrate of these environments is different? Please be more specific and clarify your point.

We now do a better job of characterizing environmental differences between sampling sites. See explanation above. 

342 – 346 – This is the first mention that there are no reproductive barriers to gene flow within your study river system. This should come much earlier, or if you do mention it earlier, should be emphasized, because it is important to your findings, specifically that morphological differences might not have a distinct genetic basis and therefore would be the result of phenotypic plasticity.

We now discuss the presence of lack of reproductive barriers in our Methods section, as well as the discussion. We agrees that this provides strong support for our conclusion that what we see is likely the result of phenotypic plasticity. See lines 115-117. 

Thank you for the opportunity to respond to reviews. We have carefully addressed each of the suggestions, and we think the paper is much improved. 

Sincerely,

Peter Searle, for all authors

---

## [Decision Letter · Decision Letter 2]

27 Apr 2021

PONE-D-20-33010R2

Ontogenetic shape trajectory of *Trichomycterus areolatus* varies in response to water velocity environment

PLOS ONE

Dear Dr. Searle,

Thank you for submitting your manuscript to PLOS ONE. After careful consideration, we feel that it has merit but does not fully meet PLOS ONE’s publication criteria as it currently stands. Therefore, we invite you to submit a revised version of the manuscript that addresses the points raised during the review process.

I have now received one re-review and this is very positive to how you dealt with the comments and the manuscript is now recommended to be accepted. Still, this reviewer made a couple of suggestions to make the manuscript even stronger that you might want to consider before we move forward. These are optional but I return the manuscript with these suggestion whether this is something you would like to include.

We look forward to receiving your revised manuscript.

Kind regards,

Peter Eklöv

Academic Editor

PLOS ONE

Journal Requirements:

Reviewers' comments:

Reviewer's Responses to Questions

**Comments to the Author**

1. If the authors have adequately addressed your comments raised in a previous round of review and you feel that this manuscript is now acceptable for publication, you may indicate that here to bypass the “Comments to the Author” section, enter your conflict of interest statement in the “Confidential to Editor” section, and submit your "Accept" recommendation.

Reviewer #1: All comments have been addressed

2. Is the manuscript technically sound, and do the data support the conclusions?

Reviewer #1: Yes

3. Has the statistical analysis been performed appropriately and rigorously? 

Reviewer #1: Yes

4. Have the authors made all data underlying the findings in their manuscript fully available?

Reviewer #1: Yes

5. Is the manuscript presented in an intelligible fashion and written in standard English?

Reviewer #1: Yes

6. Review Comments to the Author

Reviewer #1: The authors addressed all concerns and the manuscript is easy to interpret and the take home message is clear. Two minor points: 1) do you have flow data on maximum velocity for the three different sites? I wonder if this would be interesting to include because it could be that certain flooding events (high flow) could be a major selective pressure, as opposed to average velocity (though the data suggest average velocity is enough to distinguish among the groups). 2) For the discussion, you might consider starting with your second paragraph and largely deleting the first paragraph. Perhaps you can use some of the language in the first paragraph to fill in what's missing if you start with the second, but the second is where the teeth of your results are reiterated and this makes for a stronger opening to the discussion. Otherwise, I have no major comments and am grateful for your persistence and time spent improving the manuscript. It is great!

7. PLOS authors have the option to publish the peer review history of their article (what does this mean?). If published, this will include your full peer review and any attached files.

Reviewer #1: No

---

## [Author Response · Author response to Decision Letter 2]

20 May 2021

Review: Ontogenetic shape trajectory of Trichomycterus areolatus varies in response to water velocity environment

Reviewer #1: The authors addressed all concerns and the manuscript is easy to interpret and the take home message is clear. Two minor points: 

1) do you have flow data on maximum velocity for the three different sites? I wonder if this would be interesting to include because it could be that certain flooding events (high flow) could be a major selective pressure, as opposed to average velocity (though the data suggest average velocity is enough to distinguish among the groups). 

We agree that maximum velocity would be interesting to evaluate. However, we do not have a continuous measure that could capture this, because velocities were taken as point measures when we were in the field. In addition, we agree that average velocity distinguishes among the groups fairly well.

2) For the discussion, you might consider starting with your second paragraph and largely deleting the first paragraph. Perhaps you can use some of the language in the first paragraph to fill in what's missing if you start with the second, but the second is where the teeth of your results are reiterated and this makes for a stronger opening to the discussion.

We agree that our first paragraph in the discussion is largely not needed. We made the second paragraph our lead paragraph. We included a modified sentence from our first paragraph as the beginning sentence for our second paragraph (See line 347-348). This resulted in the removal of two citations (88 and 89). These citations were not critical citations. 

Otherwise, I have no major comments and am grateful for your persistence and time spent improving the manuscript. It is great!

We appreciate the insightful comments provided during the revision of this manuscript. We are grateful for your patience as we worked through your suggestions. Thank you for helping us significantly improve our manuscript. 

Thank you for the opportunity to respond to reviews. We have carefully addressed each of the suggestions, and we think the paper is much improved. 

Sincerely,

Peter Searle, for all authors

---

## [Editor Report · Decision Letter 3]

24 May 2021

Ontogenetic shape trajectory of *Trichomycterus areolatus* varies in response to water velocity environment

PONE-D-20-33010R3

Dear Dr. Searle,

We’re pleased to inform you that your manuscript has been judged scientifically suitable for publication and will be formally accepted for publication once it meets all outstanding technical requirements.

Kind regards,

Peter Eklöv

Academic Editor

PLOS ONE
---

## [Editor Report · Acceptance letter]

3 Jun 2021

PONE-D-20-33010R3 

Ontogenetic shape trajectory of *Trichomycterus areolatus* varies in response to water velocity environment 

Dear Dr. Searle:

I'm pleased to inform you that your manuscript has been deemed suitable for publication in PLOS ONE. Congratulations! Your manuscript is now with our production department. 

Kind regards, 

on behalf of

Dr. Peter Eklöv 

Academic Editor

PLOS ONE